# Early Dynamics of Quantitative *SEPT9* and *SHOX2* Methylation in Circulating Cell-Free Plasma DNA during Prostate Biopsy for Prostate Cancer Diagnosis

**DOI:** 10.3390/cancers14184355

**Published:** 2022-09-07

**Authors:** Philipp Krausewitz, Niklas Kluemper, Ayk-Peter Richter, Thomas Büttner, Glen Kristiansen, Manuel Ritter, Jörg Ellinger

**Affiliations:** 1Department of Urology and Pediatric Urology, University Hospital Bonn, 53127 Bonn, Germany; 2Institute of Pathology, University Hospital Bonn, 53127 Bonn, Germany

**Keywords:** prostate cancer, biomarker, methylation, circulating cell-free DNA, *SHOX2*, *SEPT9*

## Abstract

**Simple Summary:**

The diagnosis and risk stratification of prostate cancer (PCa) is associated with difficulties for both the examiner and the patient: the tumor burden and the aggressiveness of the tumor should be determined as accurately as possible without putting too much psychological and physical strain on the patient. The aim of the present proof-of-concept study was to investigate the potential and dynamics of quantitative *SEPT9* and *SHOX2* methylation, two promising validated non-invasive pan-cancer biomarkers, in PCa patient tissue and circulating cell-free DNA (ccfDNA) during prostate biopsy as a diagnostic tool. *SEPT9* and *SHOX2* were hypermethylated in PCa tissue and allowed discrimination of disease status, tumor stage and grade. In ccfDNA, methylation of both biomarkers allowed discrimination of localized and metastatic disease and increased shortly after prostate biopsy only in PCa patients. Hence, *mSEPT9* and *mSHOX2* are promising non-invasive measure in PCa evaluation.

**Abstract:**

Background: The methylation status of *Septin 9* (*SEPT9*) and *short stature homeobox 2* (*SHOX2*) in circulating cell-free DNA (ccfDNA) are validated pan-cancer biomarkers. The present proof-of-concept study aimed to investigate the potential and dynamics of quantitative *SEPT9* and *SHOX2* methylation in prostate cancer (PCa) patient tissue and ccfDNA during prostate biopsy as a diagnostic tool. Methods: The methylation patterns of *SEPT9* and *SHOX2* in prostate tissue were analyzed using The Cancer Genome Atlas data set (n = 498 PCa and n = 50 normal adjacent prostate tissue (NAT)). Next, dynamic changes of ccfDNA methylation were quantified in prospectively enrolled patients undergoing prostate biopsy (n = 72), local treatment for PCa (n = 7; radical prostatectomy and radiotherapy) as well as systemic treatment for PCa (n = 6; chemotherapy and 177-Lu-PSMA-therapy). Biomarker levels were correlated with clinicopathological parameters. Results: *SEPT9* and *SHOX2* were hypermethylated in PCa tissue (*p* < 0.001) and allowed discrimination of PCa and non-tumor prostate tissue (*mSEPT9*: AUC 0.87, 95%CI [0.82–0.92]; *mSHOX2*: AUC 0.89, 95%CI 0.84–0.94). *SHOX2* methylation and mRNA levels were significantly higher in PCa tissue and increased with tumor stage and grade, as well as in patients suffering from biochemical recurrence following radical prostatectomy. *SEPT9* and *SHOX2* ccfDNA methylation allowed distinguishing patients with localized and metastatic disease (*p* < 0.001 for both). In addition, methylation levels increased shortly after prostate biopsy only in patients with PCa (Δ*mSEPT9*: *p* < 0.001 and Δ*mSHOX2*: *p* = 0.001). Conclusions: The early dynamics of methylated *SEPT9* and *SHOX2* in ccfDNA allow differentiation between PCa patients and patients without PCa and is a promising marker for tumor monitoring in the metastatic stage to determine tumor burden under systemic therapy.

## 1. Introduction

The diagnosis and treatment of prostate cancer (PCa) affects patients’ quality of life and prognosis [1]. Diagnostic evaluation includes invasive procedures like prostate biopsy and cross-sectional imaging. Despite the pain, radiation exposure and infection risks caused by conventional procedures, sufficient assessment of tumor burden and grading is not achieved [2,3,4]. One reason for this is the inter- and intratumoral heterogeneity of PCa, which is also an important rationale for multiple biopsies of the prostate. However, the indolent course of the majority of PCa, adverse effects and advancing individualized PCa treatment strategies have generated a debate regarding the utility of alternative, non-invasive measures for reliable risk stratification, for example liquid-biopsies. Currently, there are indications of a possible diagnostic benefit of blood-based tests in localized prostate cancer, but all of these rely on cost-intensive circulating tumor cell (CTC) analysis and have yet not been validated in clinical use [5,6,7].

On the other hand, less expansive non-invasive biomarkers like circulating cell-free DNA (ccfDNA) could potentially improve the diagnostic accuracy of our current tools in PCa evaluation. ccfDNA is released by tumor-associated processes into the blood and cancer-specific epigenetic changes, for example aberrant methylation patterns, enable the detection of tumor-derived DNA [8,9]. The biomarkers *short stature homeobox 2* (*SHOX2*) and *Septin 9* (*SEPT9*) are well characterized in patients with different cancer entities [10]. Furthermore, hypermethylated *SHOX2*- and *SEPT9*- (*mSHOX2* and *mSEPT9*) genes in ccfDNA were proclaimed as conclusive and powerful pan-cancer biomarkers [11]. Previous translational research has highlighted the high diagnostic accuracy of both *mSHOX2* and *mSEPT9*, recently leading to approved diagnostic biomarkers for lung (*mSHOX2*; “Epi proLung”, [12]) and colorectal cancer (*mSEPT9*; “Epi proColon”) [13]). Methylations of the DNA target regions of *SHOX2* and *SEPT9* appear to arise in the context of entity-independent tumorigenic processes and thus are potentially found in PCa. However, knowledge of the occurrence of *mSHOX2* and *mSEPT9* in PCa and the diagnostic ability of these genetic alterations is limited. Vos et al. found differences in *mSHOX2* and *mSEPT9* levels that made it possible to distinguish between healthy and tumorous tissue [11]. Whereas others demonstrated promising preliminary data for *mSEPT9* as a prognostic marker in advanced PCa [14].

The present proof-of-concept study aimed to investigate the potential and dynamics of quantitative *SEPT9* and *SHOX2* methylation in PCa tissue and ccfDNA during prostate biopsy as a diagnostic tool. The investigation was based on the assumption that ccfDNA can be detected during PCA growth and especially during prostate biopsy, as iatrogenic mechanical tissue damage may lead to tumor DNA release into the blood. Clinically established assays have been used for quantitative measures of *mSHOX2* and *mSEPT9* to increase reproducibility and to enable a possible clinical application in the near future. Apart from exploring the ability to distinguish between healthy and PCa-patients by longitudinal sampling during prostate biopsy, we also assessed time- and therapy-dependent changes of *mSHOX2*- and *mSEPT9*-ccfDNA during different disease stages.

## 2. Materials and Methods

### 2.1. TCGA (Tissue) Cohort

Intratumoral PCa methylation and mRNA patterns of *SEPT9* and *SHOX2* were investigated using data from The Cancer Genome Atlas (TCGA) Research Network (TCGA; http://cancergenome.nih.gov (accessed on 1 September 2022)); data was downloaded using the UCSC Xena Browser (https://xenabrowser.net/heatmap/, downloaded 12 October 2020). The beads cg12783819 (*SEPT9*, chromosome 17:77,373,527–77,373, GRCh38.p13 (Genome Reference Consortium Human Build 38); http://www.ensembl.org (accessed on 1 September 2022)) and cg12993163 (*SHOX2*, chromosome 3:158,103,618–158,103,667, GRCh38.p13) were studied for the methylation analyses of *SHOX2* and *SEPT9* as previously described [11]. Furthermore, transcriptome sequencing data (Log2 transformed RNA-Seq v2) was downloaded. Thus, the cohort consisted of 498 men including 345 localized PCa, 78 nodal metastasis (N1), 58 biochemical recurrent diseases, 3 visceral metastasis (M1) and 50 normal adjacent tissue (NAT) samples. Information concerning nodal status was missing in 73 patients (Nx).

For comparison of NAT with PCa tissue samples, we included tissue samples from PCa patients with unknown lymph node status. However, to improve data quality, we excluded patients with missing data points in subgroup analyses, e.g., when comparing samples from localized PCa with those from N1 PCa patients.

### 2.2. University Hospital Bonn (Plasma) Cohort

After study approval of the local Ethics Committee (vote no. 348/19), plasma samples were prospectively collected from PCa patients (52 localized PCa, 11 metastatic PCa) and healthy patients (n = 25; histopathological excluded PCa by prostate biopsy) during prostate biopsy. Sampling was performed before (T1), immediately after (T2), and one hour after biopsy (T3), see Appendix A). Further, longitudinal *mSHOX2* and *mSEPT9* ccfDNA levels were investigated during prostatectomy (n = 6), during radiotherapy (n = 1), during PSMA-lutetium therapy (n = 1) and during chemotherapy (n = 4; 3× docetaxel therapy; 1× docetaxel and subsequent cabazitaxel therapy). Blood samples were collected according to patients’ clinical availability, with a maximum of 15 blood samples per patient. All patients gave written informed consent.

Blood sampling was performed using EDTA-stabilized collection tubes. In total, 3 mL plasma were processed within 2 h. Next bisulfte conversion and DNA split were performed as described previously by Schröck et al.: Blood was collected using S-Monovette^®^ (9 mL) K3 EDTA collection tubes (Sarstedt AG & Co., Nümbrecht, Germany). The blood was centrifuged for 6 min at 1.350 *g*, and the plasma was transferred to a new tube. A second centrifugation step was performed for 6 min at 3000 *g*, and 3 mL plasma were transferred to a 15 mL centrifugation tube. Plasma was prepared within 2 h after sampling and immediately stored at −20 °C. 3 mL Silane lysis/binding buffer (viral NA, Thermo Fisher Scientific, Waltham, MA, USA) were added to 3 mL plasma and incubated for 10 min at room temperature. 65 µL Dynabeads^®^ SILANE (Thermo Fisher Scientific) and 2.2 mL ethanol (absolute, molecular biology grade) were added. The mixture was incubated 45 min at 20 rpm in a rotator. The reaction tube was transferred to a DynaMag™-15 magnet (Thermo Fisher Scientific), and the supernatant was discarded. The beads were washed with buffer I (50% [*v*/*v*] Silane lysis/binding buffer (viral NA), 50% [*v*/*v*] ethanol) and eluted with 100 µL water. A total of 150 µL ammonium bisulfite (65%, pH 5.3, TIB Chemicals AG, Mannheim, Germany) and 25 µL denaturation buffer (70 mg/mL trolox [(±)-6-hydroxy-2,5,7,8-tetramethylchromane-2- carboxylic acid] in THFA [tetrahydrofurfuryl alcohol]) were added to the eluted DNA. This reaction mixture was incubated for 45 min at 85 °C. After conversion reaction, 15 µL Dynabeads^®^ SILANE and 1 mL wash buffer I were added and incubated for 45 min at 1000 rpm and 23 °C in a thermomixer. The tube was transferred to a DynaMag™-2 magnet (Thermo Fisher Scientific), and the bound DNA was washed once with wash buffer I and three times with wash buffer II (15% [*v*/*v*] water and 85% [*v*/*v*] ethanol, absolute). Finally, the bisulfite converted DNA was eluted with 65 µL elution buffer (10 mM Tris-HCl, pH 8.0) [15]. *mSEPT9* and *mSHOX2* were quantified using a *SHOX2*/*SEPT9*/*ACTB* triplex quantitative methylation-specific real-time PCR as described in detail previously [16]. *Actin-beta* (*ACTB*) served as a reference standard and was quantified to represent the total DNA of the sample. The target regions (*SEPT9*: chromosome 17:77,373,481e77,373,540; *SHOX2*: chromosome 3:158,103,550e158,103,661; *ACTB*: chromosome 7:5532,100e5,532,228; GRCh38.p13) represent the target sequences of Epi proColon and Epi proLung tests [12,13,17]. Absolute and relative methylation levels were calculated using the ΔΔCT method, and additionally qualitative analysis (quasi-digital PCR) was performed according to Vos et al. [18]. For qualitative analysis, the number of replicates showing a positive PCR reaction was counted. Hence, a positivity score was obtained ranging from zero to six. For relative quantitative analysis, percent methylation levels were calculated for each of the six PCR replicates and the mean was computed.

### 2.3. Investigation of ccfDNA Levels during Prostate Biopsy

Investigation of dynamic and absolute quantitative changes of *SHOX2* and *SEPT9* ccfDNA methylation levels was performed in men with suspected prostate cancer (n = 72) undergoing magnetic resonance imaging (MRI)-guided prostate fusion biopsy. Suspicion of localized PCa was based on elevated Prostate-specific Antigen (PSA), abnormal digital rectal examination (DRE) and/or abnormal findings on transrectal ultrasound (US). All MRI examinations included T2-weighted sequences in transverse, coronal, and sagittal planes, diffusion-weighted imaging and dynamic contrast-enhanced T1-weighted perfusion sequences. Uro-radiologists rated and reported MRI results according to PI-RADS version 2.1 (The Prostate Imaging—Reporting and Data System Version 2.1 by the American College of Radiology, USA) [19]. The same physician performed institutional standardized systematic 12-core biopsy of the lateral peripheral parts of both prostate lobes and MRI-targeted biopsy of PCa-suspicious lesions (PI-RADS ≥ 3) in one session. Biopsies were conducted under antibiotic prophylaxis, rectal cleansing and local anesthesia. Software-assisted fusion technique (KOELIS Trinity^®^) was used. All cores were separately documented, collected and evaluated by dedicated uro-pathologists. Histopathology was conducted according to guidelines [20]. Clinically significant PCa was defined as any Gleason 3 + 4 PCa (International Society of Urological Pathology (ISUP) Grade Group 2). The number of positive cores and Gleason grade group for any cancer detected was recorded. Both, biopsy naïve and previously biopsied men were included. Blood samples were taken before (T1), immediately after (T2) and one hour after biopsy (T3), Appendix A. According to guidelines, patients with D’Amico high-risk PCa (≥cT3a or ≥ISUP 4 or PSA > 20 ng/mL) were staged by PSMA-PET/CT, patients with intermediate risk by abdominal and pelvic CT and/or bone scintigraphy [20]. The stage of the disease was determined according to the results.

### 2.4. Statistics

Statistical analysis was performed using SPSS software version 25 (SPSS Inc., Chicago, IL, USA) and GraphPad Prism Version 9.0.0, GraphPad Software, San Diego, CA USA. Mean ccfDNA methylation values, including standard deviations, median methylation range, and positivity were reported. Diagnostic accuracy was described by receiver operating characteristic curve (ROC) analysis and the area under the curve (AUCs) were computed. Mann-Whitney U assays were performed to compare *mSEPT9* and *mSHOX2* levels between PCa and healthy control samples. Correlations between ranked, ordinal data were calculated using the Spearman’s rho test and interpreted according to Cohen [21]. To correlate ordinal sizes, we used the previously published methylation positivity score by Vos et al. [18]. For linear regression analysis Pearson *p* values < 0.05 were considered statistically significant. In longitudinal assessment, statistical analyses were performed only for the biopsy cohort to avoid type II (β-) error because of the small sample size.

## 3. Results

### 3.1. TCGA (Tissue) Cohort

Patients with primary PCa (n = 498) showed intratumoral hypermethylation of both *SHOX2* and *SEPT9* (*p* < 0.001) compared to NAT (n = 50). ROC analyses demonstrated high diagnostic accuracy of *mSHOX2* and *mSEPT9* levels to determine PCa versus NAT (*mSEPT9*: AUC 0.87, 95%CI [0.82–0.92], sensitivity 75.5%, specificity 86.0%; *mSHOX2*: AUC 0.89, 95%CI [0.84–0.94], sensitivity 85.3%, specificity 84.0%; see Figure 1).

The analysis of *mSHOX2* also provided information on the extent of tumor burden and grading: *mSHOX2* levels increased with advanced ISUP grading group (*p* = 0.007). *mSHOX2* was higher in patients with biochemical recurrence following radical prostatectomy (*p* = 0.002) and men suffering from lymph node metastasis (*p* = 0.023). Furthermore, locally advanced tumor stages 3 and 4 were associated with elevated methylation levels compared to PCa that was found only in the prostate (*p* < 0.001; Appendix A).

*SHOX2* mRNA concentrations were significantly different in PCa compared to NAT (*p* < 0.001) and provided moderate discriminative accuracy (AUC 0.74, 95%CI [0.66–0.81], sensitivity 80.3%, specificity 63.5%). Interestingly, *SHOX2* expression increased with tumor burden (*p* < 0.001 for both nodal metastasis and locally advanced disease) and grade (ISUP grading group *p* < 0.001; Appendix A). *SHOX2*-mRNA and -methylation levels were moderately correlated (r = 0.43). A significant decreased *SEPT9* mRNA expression was observed in PCa compared to NAT samples (*p* = 0.026). However, this difference was of limited diagnostic potential (AUC 0.59, 95%CI [0.53–0.65], sensitivity 19.7%, specificity 88.5%). *SEPT9* mRNA expression was neither correlated with PCa stage nor grade.

### 3.2. University Hospital Bonn (Plasma) Cohort

Overall cancer detection rate was 65.3%. Details of patients’ characteristics are shown in Table 1.

Data is presented as means ± standard deviations.

Re-Biopsy, Repeated biopsy; PCa, Prostate cancer; csPCA, clinically significant prostate cancer defined as Gleason ≥ 3 + 4; PSA, prostate specific antigen; DRE, digital rectal examination; US, transrectal ultrasound; PI-RADS, The Prostate Imaging—Reporting and Data System Version 2 (PI-RADS™ v2.1).

Baseline *mSHOX2*- and *mSEPT9*-ccfDNA levels were moderately increased in men with PCa compared to healthy men; however, this finding did not reach statistical significance (*p* = 0.28 and 0.18). Nevertheless, *mSHOX2* and *mSEPT9* ccfDNA levels were significantly increased in men with metastatic PCa compared to localized PCa (both *p* < 0.001, Figure 2).

In longitudinal analysis, we found dynamic changes in *SHOX2* and *SEPT9* ccfDNA methylation during prostate biopsy, radical prostatectomy, radiation and chemotherapy, Figure 3 and Figure 5. In men undergoing prostate biopsy, a non-significant increase of *mSHOX2* (*p* = 0.22) and *mSEPT9* (*p* = 0.26) ccfDNA levels in PCa patients compared to healthy men was observed before prostate biopsy (T1). Immediately after biopsy (T2), differences between the groups increased (*mSHOX2 p* = 0.033; *mSEPT9 p* = 0.001) and further raised one hour after biopsy (T3) (*mSHOX2* and *mSEPT9*, *both p* < 0.001, Figure 3).

During the course of prostate biopsy (T1–T3) methylation levels correlated with the established non-invasive PCa biomarkers PSA and PSA-density (all *p* < 0.05 and r ≥ 0.312). Clinical PCa surrogate markers like abnormal DRE, abnormal US, and higher PI-RADS category correlated moderately with *mSEPT9* levels at T1-3. For *mSHOX2*, statistical significance between methylation status and these clinical parameters could solely be identified for T3 (see Appendix A). Moreover, we found positive correlations between *mSHOX2* and *mSEPT9* one hour after biopsy (T3) with tumor aggressiveness represented by ISUP grade (*SHOX2*: r = 0.308; *SEPT9*: r = 0.383) and tumor burden represented by the total number of tumor-bearing cores (*SHOX2*: r = 0.526; *SEPT9*: r = 0.568) and MRI index lesion size (*SHOX2*: r = 0.326; *SEPT9*: r = 0.394; see Appendix A). In addition, we performed a comparative analysis of the diagnostic accuracy of *mSHOX2* and *mSEPT9* levels one hour after biopsy and the PI-RADS score to evaluate the potential added benefit of biomarker measurement at prostate biopsy. The composite score of *mSHOX2*/*mSEPT9*/PI-RADS resulted in a slight increase in diagnostic significance in terms of PCa detection compared with the current standard of care (AUC 0.88, 95%CI [0.79–0.96], sensitivity 77.8%, specificity 96.2%; PI-RADS: AUC 0.85, 95%CI [0.76–0.94], sensitivity 88.9%, specificity 65.4%; see Figure 4).

Descriptive analysis of the prostatectomy case studies (n = 4) showed an increase of *mSEPT9* and *mSHOX2* ccfDNA in the first four hours after surgery (mean increment rise from the start) of 299.48% and 4311.87%, respectively. After 36 h methylation level approximated baseline values. In men who received systemic palliative treatment for metastatic PCa (chemotherapy or PSMA-lutetium therapy) we observed longitudinal *SHOX2* and *SEPT9* ccfDNA methylation in concordance with disease burden (see Figure 5).

## 4. Discussion

Fear and distress in men undergoing evaluation for PCa are frequent and affect their quality of life [1]. Hence, improved diagnostics by biomarkers used in place or in conjunction with conventional procedures is a desirable objective. We investigated for the first time the potential of the promising pan-cancer biomarkers *mSEPT9* and *mSHOX2* in ccfDNA during PCa diagnostic processes in the clinical setting.

We identified a significant hypermethylation of both *SEPT9* and *SHOX2* in primary prostate cancer compared to NAT with an excellent diagnostic accuracy. Hence, *mSEPT9* and *mSHOX2* could serve as biomarkers in conjunction with histopathology to improve risk stratification in controversial cases. A conceivable application is the evaluation of repeat biopsy in the diagnostic dilemma of unclear histopathological changes with malignant potential. Such changes are associated with tumor detection in about 30–40% of cases in the subsequent biopsy [22]. This argument is in line with other series demonstrating the use of methylation status analysis in tissue samples to provide an improved risk stratification for clinically significant carcinoma after unsuspicious initial biopsy [23].

Moreover, *mSHOX2* status in tissue samples was associated significantly with worse prognostic factors of PCa; inter alia, increased tumor grading, increased tumor burden and increased risk of biochemical recurrence. Previous studies already identified the methylation status of *SEPT9* as prognostic in advanced PCa [14]. Therefore, our results pave the way for further investigations of *mSHOX2* and *mSEPT9* as prognostic biomarkers, for example additive risk stratification after radical prostatectomy.

Another finding of our tissue-based analysis is that DNA methylation of *SHOX2* in PCa leads to an overexpression of *SHOX2* mRNA implying a regulation by epigenetic alterations. Hereby, mRNA expression was significantly increased with tumor grading and tumor burden. This is in line with previous findings designating *SHOX2* as an oncogene [24].

On the other hand, our study demonstrates that these promising tissue-based findings cannot be easily transferred to a blood-based examination in clinical use. The static single measurement of the methylation status of both *SEPT9* and *SHOX2* showed a strong trend for hypermethylation in PCa patients but did not reliably distinguish between healthy individuals and men suffering from PCa. Nevertheless, our results indicate *mSHOX2* and *mSEPT9* in ccfDNA as promising candidates for improved blood-based non-invasive diagnostics in PCa evaluation.

*mSEPT9* and *mSHOX2* ccfDNA level analysis allowed for the discrimination of localized and metastatic disease and offer an additional opportunity for disease monitoring in PCa. This finding is of special interest for the group of patients who present with “PSA-negative” metastatic disease. In such rare cases, due to the lack of non-invasive surrogate markers of tumor burden, diagnosis is delayed, and management is compromised [25]. In addition, we confirm other series showing the utility of quantitative *mSEPT9* and *mSHOX2* analysis in plasma for monitoring PCa response to chemotherapy on a case-by-case basis [11].

Of particular interest, we observed dynamic changes in *mSEPT9* and *SHOX2* during the course of prostate biopsy, prostatectomy, and oncologic treatment of metastatic PCa, reflecting the value of these methylation markers as surrogates for tumor mass. In a proof-of-concept experiment, we measured a sharp increase in *mSHOX2* and *mSEPT9* in ccfDNA plasma immediately after prostatectomy, with complete clearance 36 h after surgery. It is known that the half-life of ccfDNA is only a few hours, so our assay only measures ccfDNA from PCa tumor cells released intraoperatively by surgical trauma. Further, dynamic measurements of *mSEPT9* and *mSHOX2* after prostate biopsy showed an increased diagnostic value to predict PCa and thus provides the opportunity for a post-biopsy analysis. Following the idea of the validated Confirm MDX^®^ test, a selective application of blood-based biomarker analysis is conceivable in patients with ongoing suspicion of PCa after an initial negative biopsy. This would especially apply to patients who clinically have a high suspicion of PCa, because they presented with abnormal findings on DRE, US, or MRI. In this scenario, the level of *mSEPT9* and *mSHOX2* in ccfDNA could help to set both, the patients and the physicians, on the right track to either follow-up or repeated biopsy. Furthermore, it can be argued that a digital rectal massage may be necessary prior to test collection at T1 to improve the ability of discrimination concerning disease status. This additive performance before blood collection is considered to result in a release of tumor material into the periphery, as it is common practice in other validated commercial urine-based tests [26]. Furthermore, *mSHOX2*/*mSEPT9* analysis can be performed with the commercially available assays (Epi proColon and Epi proLung [12,13]) using corresponding target sequences, which can be purchased at costs in the range of other methylation marker-based assays (~$200 each).

The demonstrated increase of methylation levels of *SEPT9* and *SHOX2* in ccfDNA after tissue manipulation during prostate biopsy and radical prostatectomy in PCa patients consequently represents an interesting, remarkable biological process, as it indicates the release of tumorous cell components in peripheral blood. The correlation with local tumor burden, namely the number of tumors bearing-cores and MRI lesion size, suggests that the mechanical trauma led to the dissemination of tumorous DNA. Our results corroborate a recent series that demonstrated a biopsy-associated release of prostatic tumor cells by circulating tumor cells (CTC) analysis [5]. Joose et al. discussed whether an increased tumor aggressiveness leads to less tight cell adhesions and hence, increased CTC release and a worse progression-free survival. We also found a significantly increased amount of *mSEPT9* and *mSHOX2* after biopsy in patients with increased tumor aggressiveness represented by higher ISUP grading in accordance. On the contrary, our single case observations from patients undergoing radical prostatectomy (n = 4) showing an approximation to baseline values after 36 h preclude the idea of disease progression caused by tumorous cell components. Hence, those findings do not allow any conclusions to be drawn about possible tumor seeding risks.

Our study is not without limitations. First, the application of methylated ccfDNA as a tumor biomarker is challenging as ccfDNA concentrations in body fluids are usually in the range of single genome copies. This is especially true in patients with localized PCa, where a significantly slower process of cell turnover has to be expected compared to colorectal or lung cancer. However, the current study used a highly robust, validated and inexpensive quantitative real-time PCR test targeting the identical CpG sites for *SEPT9* and *SHOX2* as clinically approved tests [12,13]. Therefore, our data is comparable to previous findings and offer the potential to transfer preclinical findings into clinical use. Second, we can only state observational findings rather than explanations for the underlying mechanisms of action resulting in hypermethylation *ofSEPT9* and *SHOX2* in PCa patients. Consequently, our analyses of the NAT samples must be viewed with caution as we currently cannot exclude hypermethylation of SHOX2 and SEPT9 in inflammatory or benign hyperplastic processes and only a limited number of samples were evaluable. Nevertheless, we found significant differences between NAT and PCa samples, which underlines the value of both biomarkers for the differentiation between “healthy” and malignant tissue. In addition, our results on tissue-based analyses are limited by the fact that lymph node status was unknown in 15% of patients and that conclusions are limited because of unequal sample sizes in the comparison groups (n_N1_ = 78 vs. n_N0_ = 345). However, we also detected significant differences in methylation levels in ccfDNA for both biomarkers between localized and metastatic disease in the University Hospital Bonn cohort (*p* < 0.001), suggesting a high potential of *mSHOX2* and *mSEPT9* for tumor stage analysis and monitoring.

Despite this, we believe that our study provided clinically relevant findings concerning the utility of methylation analysis of the promising pan-cancer biomarkers *SEPT9* and *SHOX2* in the diagnostic setup of PCa. Especially, no other study showing dynamic changes of methylation status at different steps of PCa evaluation in clinical routine is available.

## 5. Conclusions

Minimally invasive determination of the pan-cancer biomarkers *mSHOX2* and *mSEPT9* also has great diagnostic potential in PCa. The current methylation analysis in tissue and blood showed hypermethylation of both cancer biomarkers in patients with PCa compared to patients without PCa, with methylation status of *SHOX2* and *SEPT9* allowing differentiation between healthy and diseased patients and determination of tumor burden and aggressiveness. Furthermore, we showed the diagnostic utility of quantitative, dynamic analysis of *mSHOX2* and *mSEPT9* in ccfDNA after prostate biopsy to detect PCa and confirmed the biologically exciting mechanism of tumor DNA release into the blood after mechanical manipulation of the tumor.

## Figures and Tables

**Figure 1 cancers-14-04355-f001:**
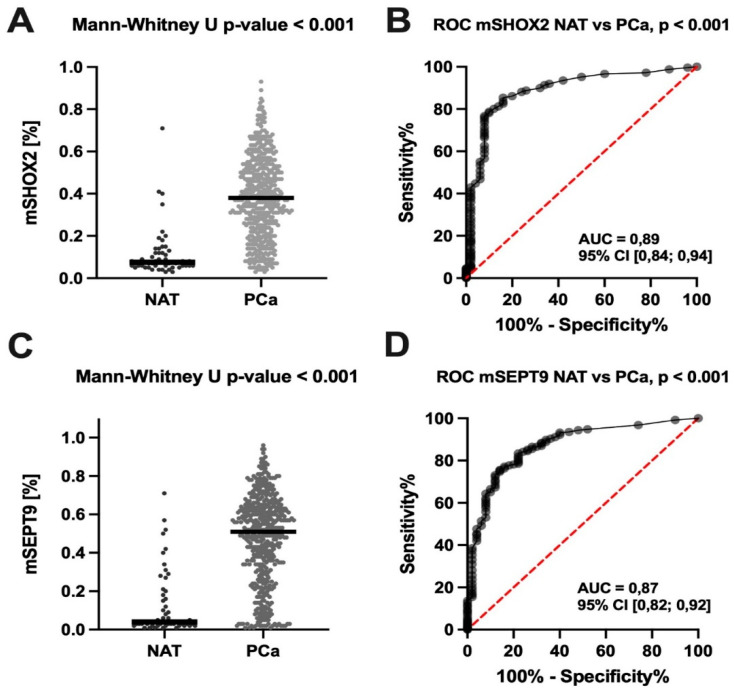
ROC curve analysis of *mSHOX2* and *mSEPT9* comparing healthy and primary PCa tissue samples. (**A**,**C**) show quantitative differences of *mSHOX2 (methylated short stature homeobox 2) and mSEPT9 (methylated Septin 9)* in NAT (normal adjacent tissue, (n = 50)) and PCa (primary prostate cancer (n = 498)). (**B**,**D**) illustrate the corresponding AUC (area under the curve) by ROC (Receiver operating characteristic) curve analysis, respectively.

**Figure 2 cancers-14-04355-f002:**
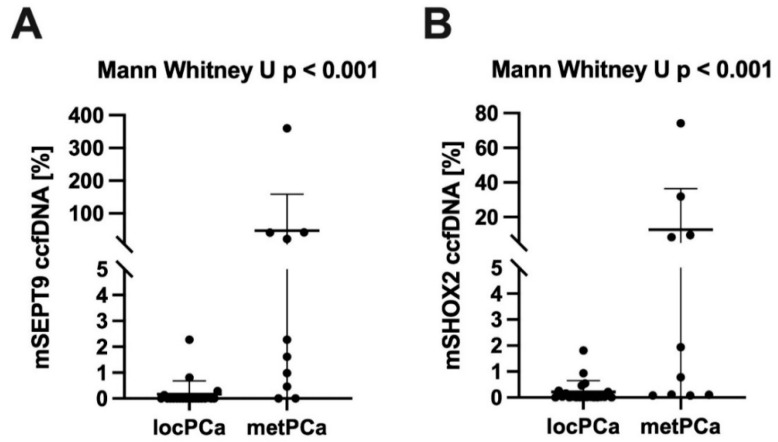
University Hospital Bonn Cohort: Quantitative methylated *SEPT9* and *SHOX2* in ccfDNA depending on tumor burden. (**A**) represents quantitative differences of *mSEPT9 (methylated Septin 9*) and (**B**) of *mSHOX2 (methylated short stature homeobox 2*) in ccfDNA (circulating cell-free DNA) between patients with locPCa (localized prostate cancer) and metPCa (metastatic prostate cancer).

**Figure 3 cancers-14-04355-f003:**
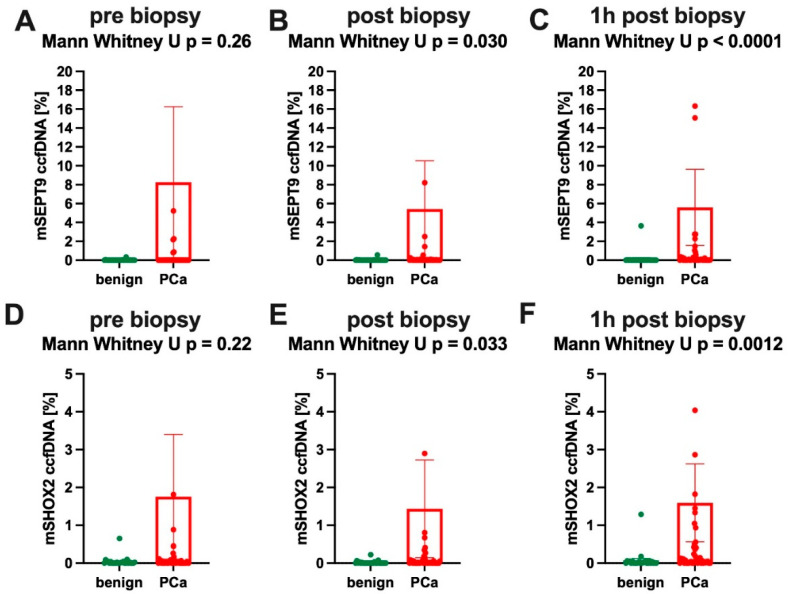
Time-dependent quantitative methylated *SEPT9* and *SHOX2* analysis during prostate biopsy. (**A**–**C**) show dynamic changes of *mSEPT9 (methylated Septin 9*) and (**D**–**F**) of *mSHOX2 (methylated short stature homeobox 2*) in ccfDNA (circulating cell-free DNA) during the course of prostate biopsy.

**Figure 4 cancers-14-04355-f004:**
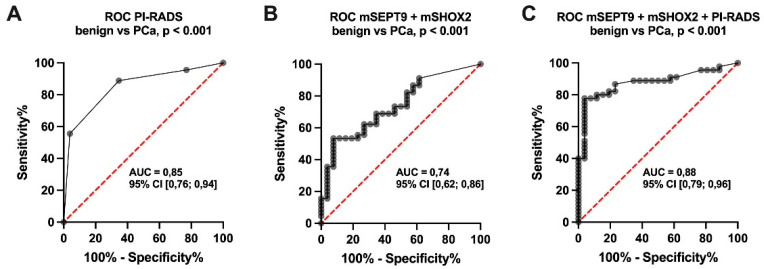
ROC curve analysis of PI-RADS vs. *mSHOX2*/*mSEPT9* one hour after biopsy and the composite score of *mSHOX2*/*mSEPT9*/PI-RADS comparing healthy and primary PCa at prostate biopsy. The ROC curve analysis of PI-RADS (**A**) vs. *mSHOX2*/*mSEPT9* (**B**) one hour after biopsy and the composite score of *mSHOX2*/*mSEPT9*/PI-RADS (**C**) comparing healthy and primary PCa at prostate biopsy is represented.

**Figure 5 cancers-14-04355-f005:**
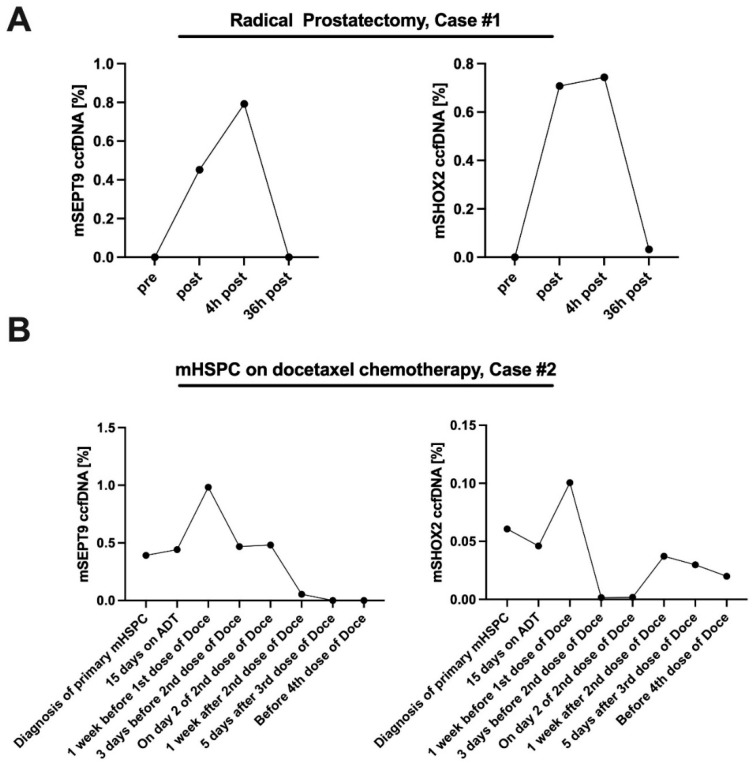
Quantitative methylated *SEPT9* and *SHOX2* ccfDNA in men undergoing radical prostatectomy and chemotherapy for metastatic PCa. The figures show representative courses of the methylation levels of *SEPT9* (*Septin 9)* and *SHOX2* (*short stature homeobox 2*) of two study patients: (**A**) in the context of radical prostatectomy and (**B**) in the context of chemotherapy with docetaxel for hormone sensitive metastatic prostate cancer (mHSPC).

**Table 1 cancers-14-04355-t001:** Descriptive statistics of University Hospital Bonn Biopsy Cohort.

	All Men (n = 72)
Age (years)	70.5 ± 7.6
PSA (ng/mL)	10.6 ± 15.7
Prostate volume (mL)	51.7 ± 27.6
Abnormal DRE (%)	43.0
Abnormal US (%)	33.3
PI-RADS < 3 (%)	16.7
PI-RADS 3 (%)	19.4
PI-RADS 4 (%)	33.3
PI-RADS 5 (%)	30.6
Re-Biopsy (%)	29.2
Total number of tumor bearing cores/patient	4.6 ± 4.9
PCa detection (%)	65.3
csPCa detection (%)	62.5

## Data Availability

The data presented in this study are available in Appendix A. Additional underlying raw data are available on request from the corresponding author.

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
