# Peer review of "Early Dynamics of Quantitative SEPT9 and SHOX2 Methylation in Circulating Cell-Free Plasma DNA during Prostate Biopsy for Prostate Cancer Diagnosis"

_cancers, 2022, doi:10.3390/cancers14184355_

Round 1

Reviewer 1 Report

Thank you for the opportunity to review the manuscript entitled "Early dynamics of quantitative SEPT9 and SHOX2 methylation in circulating cell-free plasma DNA during prostate biopsy for prostate cancer diagnosis." The authors investigate the potential of SEPT9 and SHOX2 methylation in circulating cell-free plasma DNA during prostate biopsy as a diagnostic tool. The methylation pattern was analyzed using the TCGA dataset. Dynamic changes of methylation were quantified in prospectively enrolled patients undergoing prostate biopsy (n = 72), radical treatment (n = 7), and systemic treatment (n = 6). The authors found that SEPT9 and SHOX2 were hypermethylated in prostate cancer tissue and could discriminate between prostate cancer and non-tumor prostate tissue. The authors concluded that methylated SEPT9 and SHOX2 in circulating cell-free plasma DNA allows for differentiation between patients with prostate cancer and patients without prostate cancer. It is a marker for tumor monitoring in metastatic disease to determine tumor burden during systematic therapy. The clinical topic is essential, as a correct personalized diagnosis is important for prostate cancer, the most common cancer in men, to avoid overdiagnosis and overtreatment. However, I have several comments to improve the quality of the manuscript.

  1. Could the authors conduct a comparison looking at AUCs comparing the effect of SEPT9 and SHOX9 vs. SEPT9 and SHOX9 + radiologist vs. radiologist alone to differentiate between cancerous and non-cancerous lesions. To be more robust, the authors could consider looking at radical prostatectomy specimen that provide exact ground truth data.
  2. In line 37, I suggest changing the wording from controls to patients without prostate cancer. 
  3. The abstract is missing quantitative results. It would benefit the paper if the authors included AUCs in the results section.
  4. Why are the authors not reporting specificity and sensitivity together with AUCs?
  5. What did you do about patients that had missing nodal information? Would multiple imputations be feasible? It is possible that the patients missing information on nodal status were substantially different from patients not missing this information. 
  6. What is the cost and time investment of this procedure? How does it relate to the PSA – mpMRI - biopsy pipeline? Moreover, can automated artificial intelligence methods possibly reliably and cheaply identify cancer on MRI, mitigating costs associated with the methylation analysis?   
  7. The limitations section is lacking a bit. It would benefit the paper if the authors added information on the few normal adjacent tissue samples compared to prostate cancer samples and how it affects conclusions.
  8. Please specify when reporting means and standard deviations in table 1. Moreover, could you stratify the table on whether patients had clinically significant cancer or not?

Additional minor comments:

- Could the authors spell out TCGA in the abstract?

- In the abstract, did the authors mean systemic treatment and not systematic treatment? 

- Could you add how D'Amico's high-risk prostate cancer was defined?

- Are NAT and PCa swapped in Figures 1B and 1D?

Author Response

  • Could the authors conduct a comparison looking at AUCs comparing the effect of SEPT9 and SHOX9 vs. SEPT9 and SHOX9 + radiologist vs. radiologist alone to differentiate between cancerous and non-cancerous lesions. To be more robust, the authors could consider looking at radical prostatectomy specimen that provide exact ground truth data.

Due to the reviewers comment we performed a comparative analysis of the diagnostic accuracy of mSHOX2 and mSEPT9 levels one hour after biopsy and the PI-RADS score to evaluate the potential added benefit of biomarker measurement at prostate biopsy. The composite score of mSHOX2/mSEPT9/PI-RADS resulted in a slight increase in diagnostic significance in terms of PCa detection compared with the current standard of care (AUC 0.88, 95%CI [0.79-0.96], sensitivity 77.8%, specificity 96.2%; PI-RADS: AUC 0.85, 95%CI [0.76-0.94], sensitivity 88.9%, specificity 65.4%).

We have added these findings and provided a graphical presentation of our ROC curve analysis.

  • In line 37, I suggest changing the wording from controls to patients without prostate cancer. 

We changed the wording as suggested by the reviewer.

  • The abstract is missing quantitative results. It would benefit the paper if the authors included AUCs in the results section.

Due to the valuable comment, we added quantitative results to the abstract.

  • Why are the authors not reporting specificity and sensitivity together with AUCs?

We added specificity and sensitivity values for all AUC measurements.

  • What did you do about patients that had missing nodal information? Would multiple imputations be feasible? It is possible that the patients missing information on nodal status were substantially different from patients not missing this information.

Information concerning nodal status was missing in 73 patients (Nx). Patients lacking information of nodal status were excluded from analysis.

  • What is the cost and time investment of this procedure? How does it relate to the PSA – mpMRI - biopsy pipeline? Moreover, can automated artificial intelligence methods possibly reliably and cheaply identify cancer on MRI, mitigating costs associated with the methylation analysis?

Analysis of mSHOX2/mSEPT9 can be performed with commercially available assays (Epi proColon and Epi proLung (Weiss et al. 2017; Church et al. 2014)) using corresponding target sequences, which can be purchased at costs in the range of other methylation marker-based assays (~$200 each). However, the institutional costs of testing were 5-times lower. We were able to analyze up to 15 blood samples within 8h at our institute. Therefore, a possible clinical use at reasonable costs can be postulated.

We added a sentence to outline the potential clinical use of mSHOX2/mSEPT9 analysis during prostate biopsy.

  • The limitations section is lacking a bit. It would benefit the paper if the authors added information on the few normal adjacent tissue samples compared to prostate cancer samples and how it affects conclusions.

We agree with the reviewers comment and addressed this limitation in our discussion.

  • Please specify when reporting means and standard deviations in table 1. Moreover, could you stratify the table on whether patients had clinically significant cancer or not?

We clarified the description of Table 1. Moreover, we stratified the data for csPCA and nsPCA. Only two patients were diagnosed with clinically insignificant cancer (ISUP 1). Those men were aged 64 and 53 years, biopsy naïve, had a prostate volume of 90cm³ and 38cm², a PSA of 9,9ng/ml and 1,3ng/ml, no suspicious findings on DRE and transrectal US and a PIRADS 2 and 5 rated lesion, respectively. Because of the small number of patients, we do not believe that changing Table 1 with respect to whether or not patients had clinically significant cancer would contribute to a better understanding of the data.

  • Additional minor comments:

- Could the authors spell out TCGA in the abstract?

We adjusted the wording according to the reviewers comment.

- In the abstract, did the authors mean systemic treatment and not systematic treatment? 

We adjusted the wording according to the reviewers comment.

- Could you add how D'Amico's high-risk prostate cancer was defined?

We clarified our description as follows: D'Amico high-risk PCa (≥cT3a or ≥ISUP4 or PSA > 20ng/ml).

- Are NAT and PCa swapped in Figures 1B and 1D?

We have checked our results in this regard and could not find an error.

Reviewer 2 Report

The article is devoted to an actual topic. However, the introduction to the article is partially described, the choice of markers is unclear. It is necessary to expand this part, as well as the conclusion. The choice of material for the study raises questions, since there is no unchanged normal tissue for prostate cancer. This tissue is hyperplastic, with signs of inflammation.

Author Response

  • The article is devoted to an actual topic. However, the introduction to the article is partially described, the choice of markers is unclear. It is necessary to expand this part, as well as the conclusion.

We agree with the reviewer’s assessment and clarified the choice of SEPT9 and SHOX2 as follows: Methylations of the DNA target regions of SHOX2 and SEPT9 appear to arise in the context of entity-independent tumorigenic processes and thus are potentially found in PCa. However, knowledge of the occurrence of mSHOX2 and mSEPT9 in PCa and the diagnostic ability of these genetic alterations is limited. The present proof-of-concept study aimed to investigate the potential and dynamics of quantitative SEPT9 and SHOX2 methylation in PCa tissue and ccfDNA during prostate biopsy as a diagnostic tool. The investigation was based on the assumption that ccfDNA can be detected during PCA growth and especially during prostate biopsy, as iatrogenic mechanical tissue damage may lead to tumor DNA release into the blood. Clinically established assays have been used for quantitative measures of mSHOX2 and mSEPT9 to increase reproducibility and to enable a possible clinical application in the near future.

  • The choice of material for the study raises questions, since there is no unchanged normal tissue for prostate cancer. This tissue is hyperplastic, with signs of inflammation.

We agree with the reviewers comment and state this limitation now in our manuscript: Our analyses of the NAT samples must be viewed with caution as we currently cannot exclude hypermethylation of SHOX2 and SEPT9 in inflammatory or benign hyperplastic processes and only a limited number of samples were evaluable.

Round 2

Reviewer 1 Report

·         Thanks for the opportunity to review the paper. The authors have addressed most commons sufficiently.

·    1.     More information on patients who were excluded due to missing nodal status would be interesting. Could imputations be utilized?
